# Effects of Different Pesticides on the Brewing of Wine Investigated by GC-MS-Based Metabolomics

**DOI:** 10.3390/metabo12060485

**Published:** 2022-05-27

**Authors:** Beibei Song, Yaoyao Zhou, Rong Zhan, Linjiang Zhu, Hanchi Chen, Zhi Ma, Xiaolong Chen, Yuele Lu

**Affiliations:** Institute of Fermentation Engineering, College of Biotechnology and Bioengineering, Zhejiang University of Technology, Hangzhou 310014, China; songbeibei18272807490@gmail.com (B.S.); w17757192970@gmail.com (Y.Z.); zhan15869117715@gmail.com (R.Z.); zhulinjiang@zjut.edu.cn (L.Z.); hchen23@zjut.edu.cn (H.C.); mazhi@zjut.edu.cn (Z.M.); richard_chen@zjut.edu.cn (X.C.)

**Keywords:** metabolomics, Gas Chromatographic-Mass Spectrometer(GC-MS), pesticide residue, fermentation broth, flavor substances

## Abstract

The application of pesticides is critical during the growth of high-quality grape for wine making. However, pesticide residues have significant influence on the wine flavor. In this study, gas chromatography-mass spectrometry (GC-MS) was performed and the obtained datasets were analyzed with multivariate statistical methods to investigate changes in flavor substances in wine during fermentation. The principal component analysis (PCA) score plot showed significant differences in the metabolites of wine treated with various pesticides. In trials using five pesticides (hexaconazole, difenoconazole, flutriafol, tebuconazole, and propiconazole), more than 86 metabolites were changed. Most of these metabolites were natural flavor compounds, like carbohydrates, amino acids, and short-chain fatty acids and their derivatives, which essentially define the appearance, aroma, flavor, and taste of the wine. Moreover, the five pesticides added to grape pulp exhibited different effects on the metabolic pathways, involving mainly alanine, aspartate and glutamate metabolism, butanoate metabolism, arginine, and proline metabolism. The results of this study will provide new insight into the potential impact of pesticide residues on the metabolites and sensory profile of wine during fermentation.

## 1. Introduction

Red wine has become one of the three most popular alcoholic beverages in the world because of its good flavor, special taste, and unique health benefits [1]. However, due to their high saccharide content, grapes are vulnerable to infection by pathogenic microorganisms during cultivation, such as pests, diseases, and weeds. Until now, pesticides have been widely applied to grapes to reduce potential losses [2], especially for treating grape anthracnose and grape powdery mildew [3]. The application of pesticides has also been used as an insect repellent method in more than 90% of wine-growing regions [4]. In short, grapes are most likely to be contaminated by pesticides [5]. To date, nearly 200 pesticide residues have been detected in grapes and wine industry [6].

Many works have noted that the aromatic compounds of wine are affected by pesticide residues, and the presence of pesticides consistently decreases the fermentation capability of microorganisms, including yeasts and bacteria [7]. The degradation rate of malic acid during malolactic fermentation (MLF) is reduced by about 15% in the presence of azoxystrobin, fludecinil, ciprodinil, azoxystrobin, and flutriafol during wine fermentation [8,9]. As an organic acid, malic acid has an important influence on the taste of wine. Significant changes have also been observed in the pH and color of the beer in the presence of pendimethalin, trifluralin fenitrothion, malathion, and methidathion during fermentation (*p* < 0.05) [10]. Cyanazoxazole, oxazolidone, and fluoxastrobin in the skin of grapes also decrease the contents of 2-phenylethyl acetate, ethyl butanoate, ethyl octanoate, 4-vinylguaiacol, 3-methylbutanoic acid, and methionol in wine, and these changes in volatile flavor compounds can affect the flavor of the wine [11].

In addition, pesticide residues also affect the metabolic capacity of microorganisms during wine fermentation. The presence of dichlofluanid in the skin of grapes inhibits the fermentation capability of *Leuconostoc Renos* [12]. Sieiro-Sampedro found that the fermentation activity of *Saccharomyces cerevisiae* is weakened when pesticide residues reach a certain level. The secondary metabolism of *Saccharomyces cerevisiae* can also be affected, especially in the flavor substances of wine [13]. The combination of volatile compounds in a specific ratio plays an essential role in the aroma characteristics of wine, but it is susceptible to environmental and biological factors [14]. In this case, to improve the quality of wine, it is essential to recognize the effects of pesticide residues on wine brewing.

So far, more attention has been paid to food safety issues, particularly to pesticide residues on fruits and vegetables [15]. As potent, broad-spectrum pesticides, flutriafol, hexaconazole, tebuconazole, difenoconazole, and propiconazole have the functions of protection, treatment, and eradication, exhibiting the characteristics of both a wide activity spectrum and a long-lasting effect. These pesticides are widely used on fruit trees, vegetables, and other crops to prevent pests and diseases. Recently, several studies demonstrated that the aforementioned pesticides are frequently detected in fruits and vegetables in a large amount, indicating potential risks to nontarget organisms [16]. However, only limited research has focused on the effects of pesticide residues on microbial metabolism during grape fermentation, although it may play an essential role in the quality and flavor of wine. Hence, this study has investigated the effects of five pesticides commonly used in grape cultivation on the metabolic process of wine. Structural information on the five pesticides is shown in Figure 1.

With the development of gas chromatography-mass spectrometry (GC-MS), it has been possible to realize the simultaneous detection of multiple fermentation products [17,18,19]. A powerful tool, metabolomics has been successfully applied to characterize the intermediates or end-products of the metabolism in grape and yeast samples [20]. Lee et al., revealed that metabolomics can analyze changes in metabolites in primary and secondary metabolic pathways during fermentation [21]. Furthermore, metabolomics can be used to study the internal relationship between metabolite profiles and fermentation product quality [22]. Park et al., found that metabolomics based on GC-MS is a powerful tool to study the influence of starter *Lactobacillus Plantarum* on the metabolism of amino acids, organic acids, and sugar during the fermentation of kimchi [23]. A nontargeted metabolomics approach showed that pile-fermentation was a necessary procedure for altering the polyphenols and bioactivities of Pu-erh tea [24]. Wang et al., studied the characteristic metabolite profiles and bitterness qualities during beer fermentation with metabolomics based on Ultra High Performance Liquid Chromatography- Quadrupole Time of Flight(UHPLC-Q/TOF) [25]. Metabolomics is an emerging tool for the wine industry, one that can monitor the process of wine fermentation and evaluate the flavor of wine [26]. However, reports concerning the effects and mechanism of pesticide residues on wine fermentation based on metabolomic methods have been limited. 

Taking these facts into account, we aimed to assess the effects of flutriafol, hexaconazole, tebuconazole, difenoconazole, or propiconazole treatment on the growth of *Saccharomyces cerevisiae* strains and the produced flavor substances. Metabolomic methods were applied in order to explore the changes of multiple fermentation products and metabolites during wine brewing. Our research results provide necessary information for understanding the potential action mechanism of pesticides on yeast growth and wine quality.

## 2. Result and Discussion

### 2.1. Effects of F, H, T, D, and P on the Growth of Saccharomyces cerevisiae

It is uncontroversial to state that microbiota play an essential role in the formation of aromatic compounds during wine brewing. The combined actions between the microbiota and the grape juice dynamically affect the production of the flavor components of wine. *Saccharomyces cerevisiae* is the main microbial contributor, which has been proved to regulate the aroma of wine fermentation in both positive and negative ways. Hence, to determine the effects of the five pesticides on the growth of *Saccharomyces cerevisiae* and the production of aromatic substances, *Saccharomyces cerevisiae* was exposed to the maximum residual amount of F, H, T, and D in edible grapes (0.8 mg/kg, 0.1 mg/kg, 2 mg/kg, 0.5 mg/kg). Due to the absence of a maximum residue of P on grapes, *Saccharomyces cerevisiae* was exposed to the maximum residual amount of P (0.3 mg/kg) for cowberry blueberry according to GB 2763-2019). The results are shown in Figure 2. Compared to the CK group, all five pesticides inhibited *Saccharomyces cerevisiae* growth obviously at the current amount of maximum residue (Figure 2), indicating that these pesticides may attenuate the aroma-producing capacity of *Saccharomyces cerevisiae*.

In the broadest sense, the flavor of wine refers to the complex flavor compounds formed by fermentation and aging, which is the overall impression of aroma (volatile compounds) and taste components. The final wine product is the result of various complex interactions between the grape and microbiota from the beginning of the grape winemaking process [27]. The interactions of yeast and bacteria, vines and environmental microorganisms, as well as methods of wine brewing and viticulture, affect the composition and senses of wine [28].

Yeast (mainly *Saccharomyces cerevisiae*) plays a vital role in the production of final aroma compounds during winemaking [27]. Yeast is mainly responsible for transforming grape sugars into ethanol, carbon dioxide, and other secondary but sensorial metabolites [29]. The alcohols, carbonyl compounds, phenolic compounds, fatty acid derivatives, and sulfur compounds produced by fermenting yeast are volatile, making a vital impact on the quality of the wine [30]. In short, yeast is considered the most important factor in the flavor of wine [31]. Although few studies have assessed the effect of agricultural fungicides on winemaking, data show that some pesticide residues can affect the growth of yeast cells and wine production, resulting in sluggish or ceased fermentations, affecting aroma, flavor, and overall wine quality [32]. As evidenced by Li et al., chlorothalonil can inhibit the alcoholic fermentation of Saccharomyces cerevisiae in a dose-dependent manner [33]. In addition, a strong correlation has been reported between the presence of pesticides and the occurrence of issues during alcoholic fermentation, resulting in negative impacts on the sensory quality of the final products [34]. Research on beer has shown that triadimefon negatively affects the sensory quality by changing the activity of the fermenting yeast [35]. Yeast activity during beer fermentation can be inhibited by pentamethamiphos, trifluralin, fenthion, malathion, and methion. In addition, higher amounts of residual sugars (glucose, fructose, maltose, and maltotriose) change the pH and color, as observed in all pesticide-treated samples in [36]. Moreover, interactions among microorganisms in the grape must during the fermentation process are also crucial to the quality of red wines [37]. Yeast is the dominant microorganism in the wine fermentation process, and the inhibition of its growth and metabolism by pesticides will also lead to changes in the microbial community structure of the wine, which, in turn, affects the flavor and quality of the wine [38]. In summary, the maximum residue limit of pesticides hinders the viability of yeast, influences the microbial community structure in wine, promotes the growth of several spoilage isolates, and, thus, may have a negative impact on wine aroma [39]. In our study, the five pesticides at their maximum residue limits had a significant inhibitory impact on the growth of *Saccharomyces cerevisiae*, which may further reduce the aroma-producing ability of *Saccharomyces cerevisiae* and affect wine flavor.

### 2.2. Effects of F, H, T, D, and P on Metabolites Produced in Wine Brewing

To characterize the changes of flavor and aroma compounds induced by pesticides, a metabolomics approach based on GC-MS was applied. As shown in Figure 3 and Figure 4, Principal component analysis(PCA) and Orthogonal partial least squares discriminant analysis(OPLS-DA) analyses were performed to explore the metabolic differences between the F, H, T, D, P, and CK groups according to the GC-MS data. The PCA score plot showed good separations between CK and the F, H, T, D, P-treated groups, as well as good model qualities (R^2^X = 0.627, 0.552, 0.735, 0.627 and 0.627, respectively). The results revealed that there were apparent differences in the metabolites in wines treated with different pesticides (Figure 3), including the aroma substances. Among them, the T treatment exhibited the greatest influence on the aroma substances in wine, followed by F, H, and P. D treatment exhibited the least impact, since there was an incomplete separation between the D and CK groups (Figure 3F).

For the OPLS-DA model, both the results of permutation analysis (Appendix A) and Q^2^ (0.985, 0.89, 0.981, 0.913, 0.976 in the F, H, T, D, P groups, respectively) demonstrated the good qualities and predictive abilities of these models. Significant changes in the volatile substances were induced in T, H, F, and P-treated groups during wine fermentation. According to the cutoff value of VIP >1 and *p* < 0.05, 27 increased and 8 decreased metabolites were identified in the F group in comparison to the CK group (Figure 4A). Moreover, 23 increased and 3 decreased metabolites were found in the H group (Figure 4B). Consistent with the results of the PCA, T and D treatment caused the most (41) and the least (17) changed metabolites, respectively (Figure 4C,D). In the P group, 17 metabolites were increased and 9 metabolites were decreased, compared to CK group (Figure 4E). The details of metabolites, FCs, and *p*-values for each significantly altered metabolite are shown in Appendix A. These metabolites belong in esters, acids, alcohols, phenols, alkanes, amino acids, etc., most of which have been identified as important compounds in wine aroma, and are associated with the fruity and sweet taste [37].

To make more intuitionistic analyses of the volatile pattern, heatmap and upset plots were performed for visualizing the relative changes in metabolite contents (Figure 5 and Figure 6). Heatmapping provided visual access to each chemical in the wine, as well as their relative contents. It facilitated the comparison among the F, H, T, D, P and CK groups. The chromatic scale of the heatmap indicated the relative amount of each compound (blue for the minimum, red for the maximum). According to the similarities among these compounds, hierarchical clustering analysis was performed, and two main clusters were observed between the CK and pesticide treated-groups (Figure 5). However, L-5-Oxoproline and 3-β Mannobiose were specifically changed in the H group (Figure 5B). The 2-propenamide, methoxyacetic acid, and D-(-)-Fructofuranose were specifically changed in the D group (Figure 5D). The ethanol, N- (4-butyl)acetamide, aminoacetaldehyde dimethyl acetal, and 2-aminoheptanedioic acid were specifically varied in the P group (Figure 5E). More downregulated metabolites were observed than upregulated metabolites in the F, H, T, and D groups. It was the opposite in the P group. These results demonstrated that different pesticides have specific effects on the flavor and aroma compounds in wine during winemaking.

The changed metabolites were classified among different pesticides groups (Figure 6). This revealed that two metabolites (1,4-benzenediol and arabinofuranose) were corporately changed among the five pesticide-treated groups, indicating that these compounds represented potential biomarkers to evaluate the pesticide contamination of wine. Twenty and ten metabolites related to acids and carbohydrates were exclusively changed by T and F treatment, respectively (Figure 6; Appendix A). Furthermore, eight metabolites were corporately varied by four of five pesticides, and nine metabolites were corporately disturbed by three of five pesticides (Figure 6), indicating that the effects of the five pesticides on wine flavor were similar, to a certain extent.

Overall, treatment with these pesticides mainly changed the concentrations of carbohydrates, alcohol, acids, and amino acid, which are the main contributors of wine taste and flavor. Carbohydrates are not only the key factor in wine taste, but also the main energy supplier for the growth of yeasts [40]. In our study, the arabinofuranose increased in all pesticide-treated wines, suggesting the varied taste of wine and growth of yeasts. Arabinofuranose is obtained from the hydrolysis of terpene, monosaccharide, or disaccharide by arabinofuranosidase [41]. The increased arabinofuranose induced by the pesticides may be caused by the changed activity of arabinofuranosidase, thus affecting the special aroma and flavor of wine. Glucose and fructose are the main sugars associated with the sweetness of grape berries [42], while the significantly reduced fructose content in the D group would provide the wine a bitter flavor (Figure 5D). Trehalose is an important reserve carbohydrate and a stress protector in yeast [43]. In our study, its content was significantly downregulated in the F group, which could also affect the flavor of wine, to a certain extent (Figure 5A). In recent reports, various monoterpene disaccharide glycosides, specifically, rutinosides and α-L-arabinofuranosyl-β-D-glucopyranosides, represent a novel class of metabolites in grapes and flavor compounds in wine [44]. 3-βMannobiose, as a monoterpene diglycoside, was significantly downregulated in the F and T treated groups (Figure 5C).

The organic acids in wine have an important effect on the physical and chemical balance of flavor (such as pH), thus affecting the quality of wine. In the P, T, or F groups, the main organic acids that varied in the wine were 2-aminopimelic acid, lactic acid, acetic acid, glutaric acid, citric acid, malic acid, D-gluconic acid, and oxalic acid (Figure 5A,C,E). Citric acid is a precursor for the synthesis of ethanol, isobutanol, and lactic acid by yeast, which has a great influence on the metabolic process of yeast and subsequent flavor of wine [45]. The citric acid was significantly increased in the T group, reflecting that the citric acid was useless to *Saccharomyces cerevisiae* in the presence of T (Figure 5C). Compared to other groups, the contents of malic acid in the F and T groups were significantly decreased. In the T group, concomitantly decreased lactic acid indicated that T might be an inhibitory factor of MLF in yeast (Figure 5C). In general, organic acids bring different acidity perceptions to form a harmonious sourness: lactic acid provides a soft sourness, (slightly milky in red wine), citric acid is refreshing and cool, and malic acid represents a bit of pungent sourness [46]. Furthermore, treatment with H increased the caprylic and acetic acid concentrations, which provides the wine with an unpleasant taste. On the other hand, treatment with T increase the concentration of acrylic (unpleasant) and glycolic (rancid smell) acids; acrylic acid is a pungent liquid organic acid that can affect wine aromas [1]. In groups F and H, the content of 2-Ethyl-3-hydroxypropionic acid was significantly upregulated, while propionic acid had a strong pungent odor, which affects the flavor and quality of wine. Therefore, treatment with the tested pesticides effected changes in the acid content, possibly because they reduce the synthesis of the corresponding esters during fermentation. Free amino acids can be produced by proteolysis during red wine fermentation through metabolism by endogenous enzymes or microorganisms. Microbes interact with changes in free amino acid composition. The accumulation of glutamic acid (Glu), which contributes to the freshness and umami of wine, has a trend of significant upregulation in the F, H, and P groups [47].

The production of alcohols like ethanol and 2,3-butanediol from sugar degradation by *Saccharomyces cerevisiae* under anaerobic conditions is a vital process of wine fermentation [48]. In addition to being beneficial to taste, ethanol also had a comprehensive influence on the preservation of wine, and it affects the perception of other aromatic compounds [49]. Studies have shown that the increase in sweetness is proportional to the increased amount of ethanol, while the bitterness level correspondingly decreases [50]. The content of ethanol was significantly increased after interaction with the P, T, and F groups, indicating that pesticide treatment at an early stage of wine fermentation has a certain effect on sugar and acid metabolism (Figure 5A,C,E) [51,52]. 2,3-butanediol was the second largest byproduct of alcohol fermentation, which affects the aroma and body of wine due to its bitterness and viscosity [53]. In the T and F groups, the reduced 2,3-butanediol content reflected that the pesticides induce a reduced bitterness and viscosity in the wine (Figure 5A,C). In addition, they reduce the fermentation activity of *Saccharomyces cerevisiae*, thereby affecting the synthesis of volatile compounds in other metabolic pathways.

### 2.3. Altered Metabolic Pathway by F, H, T, D and P Treatment during Wine Brewing

The mechanism of effects from pesticides on the flavor substance during wine brewing has been explored by metabolic pathway analysis using the Kyoto Encyclopedia of Genes and Genomes(KEGG) database (Figure 7 and Figure 8). In the F, H, T, and P groups, the most significantly changed pathways occurred in alanine, aspartate and glutamate metabolism, butanoate metabolism, arginine, and proline metabolism (Figure 7A–C,E). D treatment showed a major impact on arginine biosynthesis and glycerolipid metabolism (Figure 7D), which also changed in the P-treated group (Figure 7E).

Carbohydrate metabolism and amino acid metabolism are important pathways for the production of volatile flavor compounds. The amino acids in red wine come from raw materials and microbial synthesis, and some amino acids are the precursors of higher alcohols. Therefore, the biosynthesis of amino acids and higher alcohols is related to carbohydrate metabolism and amino acid metabolism. In addition, fatty acid biosynthesis is related to carbohydrate metabolism, amino acid metabolism, and lipid metabolism [54]. Amino acids (AA) are the precursors of most volatile compounds, which are also considered to affect the aroma of wine. During fermentation, many altered compounds induced by pesticides are involved in amino acid metabolism pathways. Glutamate has the potential to enhance the umami taste of beer, which was increased in wine treated with F, H, and P (Figure 7A,B,E). 4-aminobutyric acid (GABA) forms from glutamate metabolism by glutamate decarboxylase [55], which was increased significantly in the F, H, T, and P groups (Figure 7A–C,E). During wine brewing, the effects on microorganisms from pesticides also regulate the release of decarboxylase and the increase of GABA in wine [56]. Proline is the main amino acid in grapes, but it cannot be adequately utilized by *Saccharomyces cerevisiae* during wine brewing [57]. The contents of proline and lysine are related to arginine and proline metabolism and arginine biosynthesis, which were significantly downregulated in the T group (Figure 7C). Although a change in arginine was not observed, it has been reported that arginine is an inhibitory factor of proline utilization and partially represses the expression of genes involved in proline degradation in yeast [57]. Together with proline and ammonia nitrogen, free AA constitutes most of the yeast’s assimilable nitrogen (YAN), providing nutrients for yeast reproduction and growth [57]. Under fermentation conditions, *Saccharomyces cerevisiae* consumes most of the YAN (and other nutritional factors) in wine to produce various biologically active substances, especially volatile compounds [48]. Moreover, some derivatives of arginine and proline metabolism, including L-ornithine, lysine, and putrescine, also change, which is induced by these pesticides during fermentation. Lysine, which is related to the bitterness of wine, showed a significant downward trend in the T group (Figure 7C). Therefore, pesticide residues can also affect taste and flavor of wine by changing metabolic activity of yeast and the metabolism of AA.

In wine, lipids are derived from grape skins, pulp, seeds, and yeast cell walls. Lipids play an important role during winemaking by facilitating the penetration of amino acids into yeast, regulating yeast metabolism, and restricting excess production of acetic acid. Studies have proved that grape fat is a “survival factor” of yeast because it fully supplies the growth of yeast and extends its fermentation activity in wine [57]. In this study, the significantly reduced glycerol content in the T group was related to glycerolipid metabolism, which further verified that pesticides affect the fermentation activity of yeast (Figure 2, Figure 7C and Figure 8). In conclusion, the enrichment analysis of metabolic pathways in red wine samples showed that fatty acid biosynthesis, carbohydrate metabolism, and amino acid metabolism play an important role in the formation of red wine flavor.

## 3. Materials and Methods

### 3.1. Chemicals and Reagents

Ethanol, methanol, isopropanol, and acetonitrile were purchased from Aladdin (Shanghai, China). Flutriafol (F), hexaconazole (H), tebuconazole (T), propiconazole (P), and difenoconazole (D) were provided by the Agro-Environmental Protection Institute, Ministry of Agriculture and Rural Affairs (Figure 1). The five pesticides were dissolved in acetone for preparing stock solutions with concentrations as follows: flutriafol (12.5 mg·mL^−1^), hexaconazole (3.5 mg·mL^−1^), tebuconazole (13.3 mg·mL^−1^), difenoconazole (6.7 mg·mL^−1^), and propiconazole (30.9 mg·mL^−1^). The active dry yeast (*Saccharomyces cerevisiae*) was bought from Angel Yeast Co. Ltd. (Hangzhou, China). All other chemicals and reagents purchased from Aladdin (Shanghai, China) were analytical grade. 

### 3.2. Experiment on the Effect of Pesticides on the Growth of Saccharomyces cerevisiae

Rich YPD medium (1% yeast extract, 2% peptone, 2% glucose, and maximum residue limit of pesticides), was used to culture *Saccharomyces cerevisiaes* and for chronological lifespan experiments. Samples were taken regularly within 48 h, and OD_600_ was detected to monitor the effects of the five pesticides on the growth of *Saccharomyces cerevisiae*.

### 3.3. Preparation of Wine Samples

Grapes after washing and drying were smashed and spiked with a certain amount of pesticide solution. Subsequently, the mixture was filtered and mixed with 20.0 mg NaHSO_4_ and 0.2 g *Saccharomyces cerevisiaes* to obtain grape juice. A total of 500 mL of grape juice was added to each fermentation bottle. There were 5 groups treated with flutriafol (F group), hexaconazole (H group), tebuconazole (T group), propiconazole (P group), and difenoconazole (D group), as well as 1 control group (CK group), with 8 parallels in each group. The final concentrations of F, H, T, P, and D in the fermentation bottle were 0.8 mg·kg^−1^, 0.1 mg·kg^−1^, 2.0 mg·kg^−1^, 0.5 mg·kg^−1^, and 0.3 mg·kg^−1^, respectively, according to the national standard (GB 2763-2019). The *Saccharomyces cerevisiaes* was precultured by sweetwater for 2 h and then inoculated into yeast extract peptone dextrose medium (YPD) in flasks for 2 days at 26 °C. The fermentation phenomenon was recorded every day.

### 3.4. Metabolite Extraction from Wine Samples

The quality control (QC) samples were prepared by mixing with all test sample extracts in equal volumes. Subsequently, 200 μL of the wine sample and QC mixture was dried with a nitrogen blower. An amount equal to 80 μL of methoxyamine pyridine hydrochloride solution (15 mg·mL^−1^) was added to each sample. After vortexing for 2 min, the sample mixtures were shaken in a vibration mill at 37 °C for 90 min. After that, 80 μL of the derivatization reagent mixture comprising BSTFA-TMCS (Bis (trimethylsilyl)trifluoroacetamide, with 1% trimethylchlorosilane) and 20 μL of n-hexane was added. The sample mixtures were shaken in a vibration mill at 70 °C for 60 min. Supernatants of the derivatized samples were transferred to sample vials for GC-MS analysis.

### 3.5. Metabolomics Analysis by GC-MS

The derivatized samples were analyzed by an Agilent 7890 A gas chromatography system coupled with an Agilent 5975 C MSD system (Agilent, Santa Clara, CA, USA). An HP-5MS fused-silica capillary column (30 m × 0.25 mm × 0.25 μm, Agilent J & W Scientific, Folsom, CA, USA) was utilized to separate the derivatives. Helium (>99.999%) was used as the carrier gas at a constant flow rate of 1.0 mL·min^−1^. The injector temperature was maintained at 260 °C. Injection volume was 1 μL by splitless mode. With a starting temperature of 50 °C, the column temperature was increased to a final temperature of 125 °C at a speed of 15 °C·min^−1^ and then raised to 210 °C at 5 °C·min^−1^, followed by 10 °C·min^−1^ to 270 °C, and 20 °C·min^−1^ to 305 °C, at which it maintained for 5 min. MS quadrupole and ion source (electron impact) were set to 150 and 230 °C. The collision energy was 70 eV. Mass data were acquired in a full-scan mode (*m*/*z* 50–450), and the solvent delay time was set to 5 min.

### 3.6. Data Processing and Multivariate Analysis

The acquired MS data from GC−MS were analyzed by ChromaTOF software (Santa Clara, CA, USA; St Joseph, MI, USA). Furthermore, metabolites were qualified by the Fiehn database, which was linked to the ChromaTOF software. Briefly, after the alignment with the Statistic Compare component, a CSV file was obtained with three-dimension datasets, including sample information, peak name, retention time, *m*/*z*, and peak intensity. The internal standard was used for data quality control (reproducibility). The internal standards and any known pseudo-positive peaks, such as peaks caused by noise, column bleed, or BSTFA derivatization procedure, were removed from the dataset. The peaks from the same metabolite were combined. The raw data were normalized to the total peak area of each sample in Excel 2007 (Microsoft, Redmond, DC, USA). 

The resulting data were imported into a SIMCA (version 14.0, Umetrics, Umeå, Sweden) in order to perform subsequent analyses, including principal component analysis (PCA), partial least-squares discriminant analysis (PLS-DA), and orthogonal partial least-squares discriminant analysis (OPLS-DA). The Hotelling’s T_2_ region, shown as an ellipse in score plots of the models, defined the 95% confidence interval of the modeled variation. The quality of the models was described by the R_2_X, R_2_Y, and Q^2^ values. R_2_X or R_2_Y was defined as the proportion of variance in the data explained by the models, indicating the goodness of fit. Q^2^ was defined as the proportion of variance in the data predicted by the model and calculated by a cross-validation procedure, indicating predictability. A default seven-round cross-validation in SIMCA was performed to determine the optimal number of principal components and avoid model overfitting. The OPLS-DA models were also validated by permutation analysis (200 times). Analysis of variance (ANOVA) and a subsequent Dunnett’s post hoc test were used to detect whether there was significant difference among the control and pesticide-treated groups. All statistical analyses were carried out in OmicStudio software (https://www.omicstudio.cn/index. accessed on 22 August 2021). According to volcano map, characters simultaneously met VIP > 1, *p*-value < 0.05 and fold change (FC > 1.5 or <2/3) were identified as significantly changed metabolites (as denoted by *). MetaboAnalyst, a free and web-based tool with a high-quality KEGG metabolic pathway, was used for pathway analysis (http://www.metaboanalyst.ca. accessed on 22 August 2021).

## 4. Conclusions

In this study, the effects of hexaconazole, difenoconazole, flutriafol, tebuconazole, and propiconazole on the growth of *Saccharomyces cerevisiae* were investigated, and the changes in the flavor substance in wine were characterized by metabolomic analysis based on GC-MS. Results revealed that the growth of *Saccharomyces cerevisiae* is significantly inhibited by these five pesticides at the maximum residue limit, and the fermentation profile of *Saccharomyces cerevisiae* was significantly changed. After being treated with the five pesticides, a total of 86 changed metabolites were detected during wine brewing in comparison to the control, and most of the metabolites were natural flavor compounds, such as carbohydrates, amino acids, short-chain fatty acids, and alcohol, which essentially define the appearance, aroma, flavor, and taste of wine. The changes in alanine, aspartate and glutamate metabolism, butanoate metabolism, arginine, and proline metabolism were the most significant metabolism pathways in the hexaconazole, difenoconazole, flutriafol, tebuconazole and propiconazole treatment groups. Difenoconazole treatment had a significant impact on arginine biosynthesis and glycerolipid metabolism. Among them, tebuconazole exhibited the greatest negative influence on the wine quality for its effects on the aroma substances, followed by flutriafol, hexaconazole, propiconazole, and difenoconazole. The results of our study comprehensively analyzed the effects of pesticides on wine flavor and provided some key information for understanding the potential function mechanisms on *Saccharomyces cerevisiae* growth and wine quality. Our study pointed out that integrating GC-MS with multi-statistical analysis can accurately assess pesticide-induced flavor differences in wine. It further proved the feasibility of metabolomic methods to broadly characterize the chemical composition of wine during fermentation, and provides a new idea for future research on the flavor quality of fermented wine.

## Figures and Tables

**Figure 1 metabolites-12-00485-f001:**
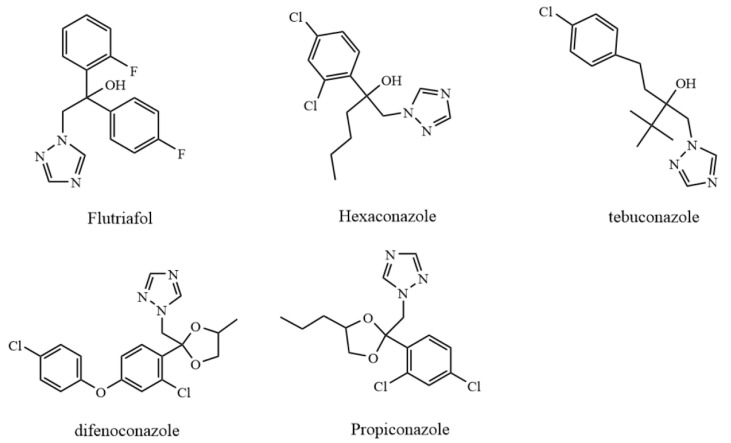
Chemical structure diagrams of five pesticides.

**Figure 2 metabolites-12-00485-f002:**
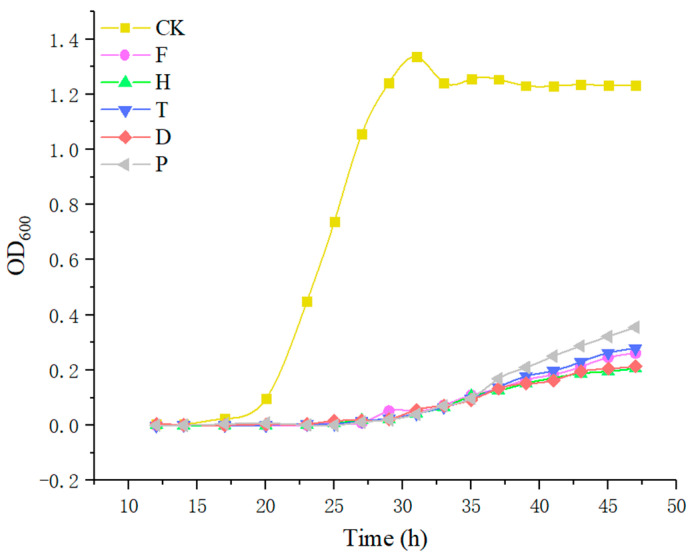
The growth curve of yeast exposed to the maximum residual amount of F, H, T, D, and P. Note: CK, blank control group; F, flutriafol; H, hexaconazole; T, tebuconazole; D, difenoconazole; P, propiconazole.

**Figure 3 metabolites-12-00485-f003:**
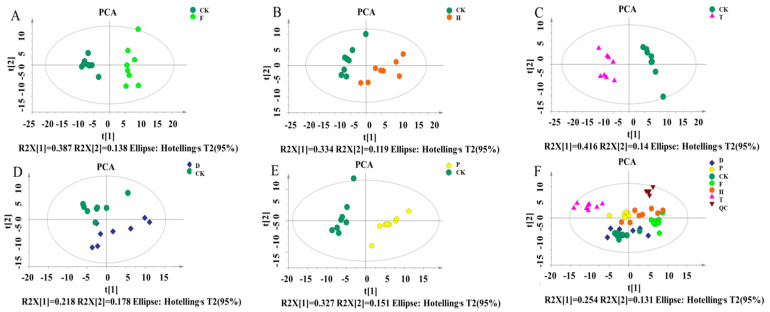
PCA diagram of different metabolites collected during wine fermentation with the treatment of different pesticides. (**A**) CK and F; (**B**) CK and H; (**C**) CK and T; (**D**) CK and D; (**E**) CK and P; (**F**) All collected data are within the 95% confidence interval. Red represents the CK group, yellow represents the F group, light blue represents the H group, purple represents the T group, green represents the D group, blue represents the P group, and light purple represents the QC group. Note: CK, blank control group; F, flutriafol; H, hexaconazole; T, tebuconazole; D, difenoconazole; P, propiconazole.

**Figure 4 metabolites-12-00485-f004:**
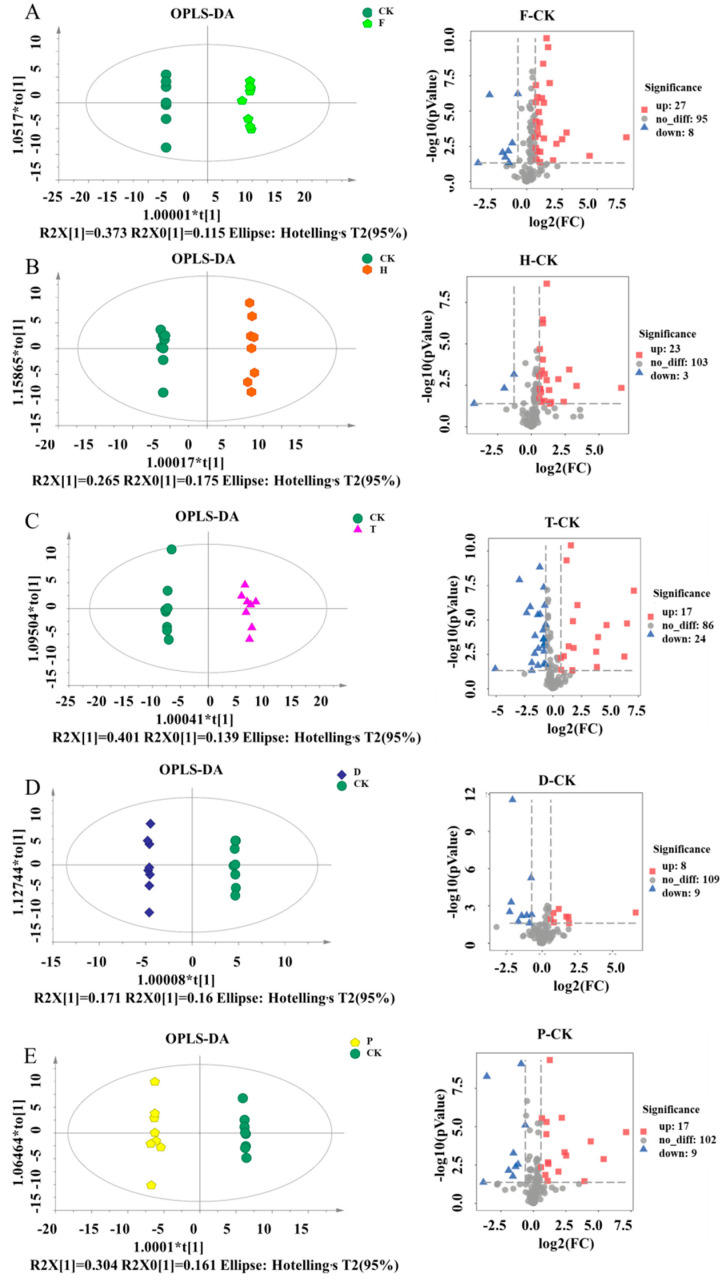
OPLS-DAand Volcano map of changed metabolites during wine brewing with the treatment of F, H, T, D, and P. (**A**) CK and F; (**B**) CK and H; (**C**) CK and T; (**D**) CK and D; (**E**) CK and P. All collected data are within the 95% confidence interval. Red represents the CK group, yellow represents the F group, light blue represents the H group, purple represents the T group, green represents the D group, blue represents the P group, and light purple represents the QC group. Note: CK, blank control group; F, flutriafol; H, hexaconazole; T, tebuconazole; D, difenoconazole; P, propiconazole.

**Figure 5 metabolites-12-00485-f005:**
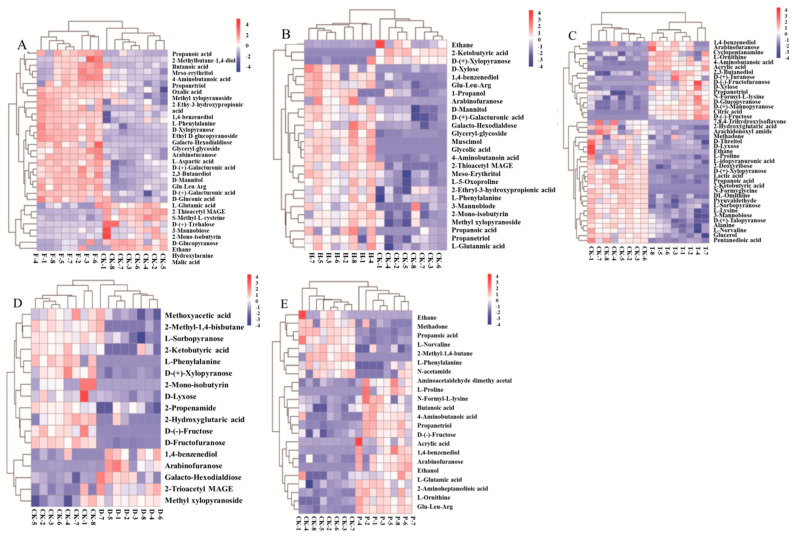
Heatmap of primary altered metabolites collected during wine brewing with the treatment of F, H, T, D, and P. (**A**) CK and F; (**B**) CK and H; (**C**) CK and T; (**D**) CK and D; (**E**) CK and P Note: F, flutriafol; H, hexaconazole; T, tebuconazole; D, difenoconazole; P, propiconazole.

**Figure 6 metabolites-12-00485-f006:**
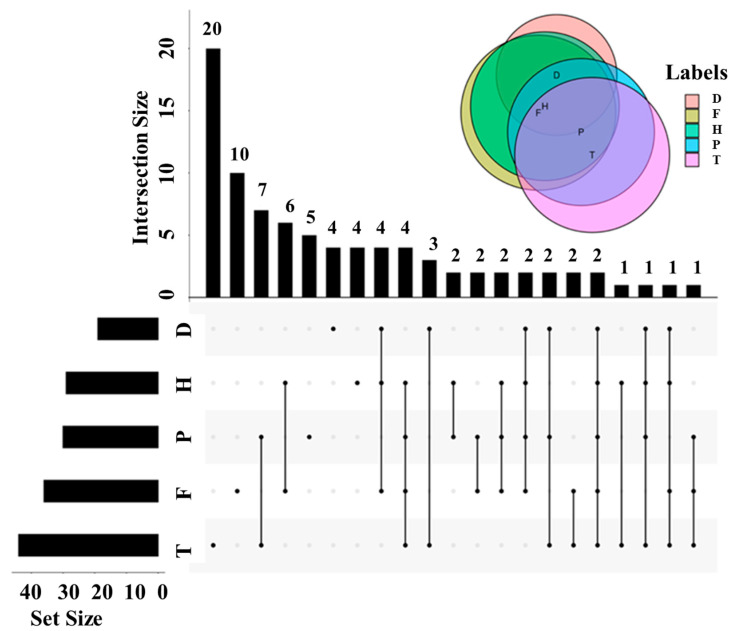
Upset plot of the changed compounds during wine brewing with the treatment of F, H, T, D, and P. Note: F, flutriafol; H, hexaconazole; T, tebuconazole; D, difenoconazole; P, propiconazole.

**Figure 7 metabolites-12-00485-f007:**
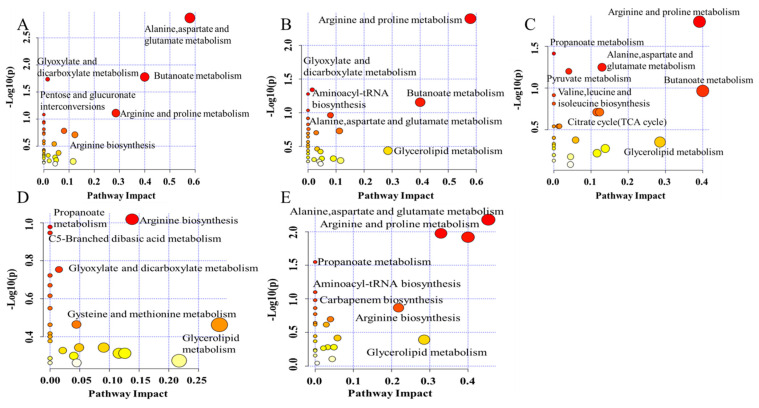
Changes in metabolic and biosynthetic pathways during fermentation with the treatment of F (**A**), H (**B**), T (**C**), D (**D**), and P (**E**). Note: CK, blank control group; F, flutriafol; H, hexaconazole; T, tebuconazole; D, difenoconazole; P, propiconazole.

**Figure 8 metabolites-12-00485-f008:**
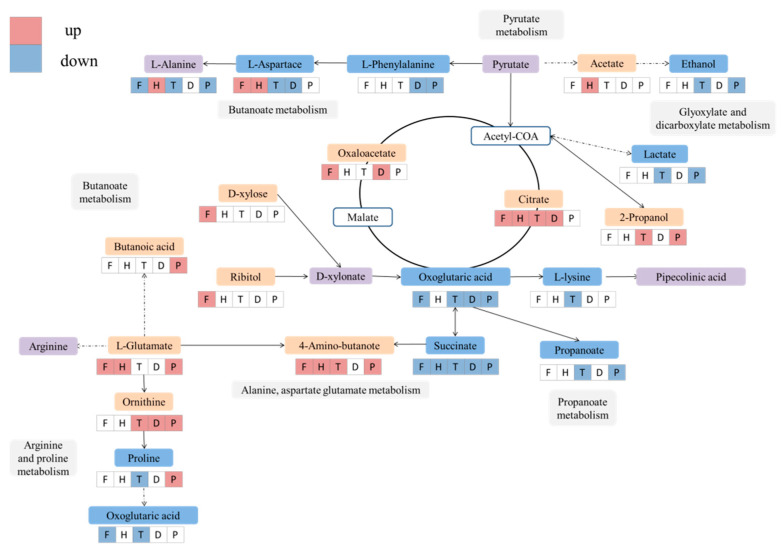
Effects on metabolic pathways of wine fermentation under the treatment of F, H, T, D, P. Note: F, flutriafol; H, hexaconazole; T, tebuconazole; D, difenoconazole; P, propiconazole.

## Data Availability

Data is contained within the article.

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
