# Peer review of "Effects of Different Pesticides on the Brewing of Wine Investigated by GC-MS-Based Metabolomics"

_metabolites, 2022, doi:10.3390/metabo12060485_

Round 1

Reviewer 1 Report

The effect of pesticides on yeast growth was not explained in the Material and methods section.

Microorganisms names should be written adequately.

Doscussion of the effect on yeast growth should be improved.

Generally, discussion shoud be more profound.

Author Response

Response to Reviewer 1 Comments

Point 1: The effect of pesticides on yeast growth was not explained in the Material and methods section.

Response 1: Thanks for your comments. The corresponding experimental part has been added in the paper as the following:

2.2. Experiment on the effect of pesticides on the growth of Saccharomyces cerevisiae

Rich YPD medium (1% yeast extract, 2% peptone, 2% glucose and maximum residue limit of pesticides), was used to culture saccharomyces cerevisiaes and for chronological life span experiments, respectively. Samples were taken regularly within 48h, and OD600 was detected to monitor the effects of five pesticides on the growth of saccharomyces cerevisiae.

Point 2: Microorganisms names should be written adequately.

Response 2: Thank you. The detailed information of the microorganisms names has been supplemented in the relevant places of this new manuscript. The active dry yeast (Saccharomyces cerevisiae) was bought from Angel Yeast Co. Ltd. (Hangzhou, China) with the comercial name of Angel wine yeast.

Point 3: Discussion of the effect on yeast growth should be improved.

Response 3: Thank you for the comments. We supplied more reasons for how pesticides affect growth of yeast in Page 6 Line 8-13 and Page 6 Line 19-25 , as follows:

Although few studies assessed the effect of agricultural fungicides on winemaking, datas show that some pesticides residues can affect the growth of yeast cells and wine production, resulting in slugged or stopped fermentations, affecting wine aroma, flavor, and overall wine quality [32]. Li et al. evidenced that chlorothalonil can inhibit the alcoholic fermentation by saccharomyces cerevisiae in a dose-dependent manner [33]. In addition, a strong correlation has been reported between the presence of pesticides and the occurrence of issues during alcoholic fermentation, resulting in negative impacts on the sensory quality of the final products [34].

Moreover, the interactions among the microorganisms in the grape must during the fermentation processes are also crucial for the quality of red wines [37]. Yeast is the dominant microorganism in the wine fermentation process, and the inhibition of its growth and metabolism by pesticides will also lead to changes in the microbial community structure in wine, which in turn affects the flavor and quality of wine [38]. In summary, maximum residue limit of pesticides hindered the viability of yeast and Influenced microbial community structure in wine, promoted the growth of several spoilage isolates and thus may have a negative impact on wine aroma [39].

Point 4: Generally, discussion shoud be more profound.

Response 4: Thanks. The corresponding discussion part has been added in Page 11 Line 17-35, as follows:

In general, organic acids bring different acidity perceptions to form a harmonious sourness: lactic acid gives a soft sourness, slightly milky to red wine, citric acid is refreshing and cool, and malic acid represents a bit of pungent sourness [46]. Treatment with H increases the caprylic and acetic acid concentrations, which give the wine an unpleasant taste. On the other hand, treatment with tebuconazole increased the concentration of acrylic (unpleasant) and glycolic (smell of rancid), and acrylic acid a pungent liquid organic acid which can affected wine aromas [1]. In groups F and H, the content of 2-Ethyl-3-hydroxypropionic acid was significantly up-regulated, while propionic acid had a strong pungent odor, which affected the flavor and quality of wine. Therefore, treatment with the tested pesticides affected changes in acid content, possibly because they reduced the synthesis of the corresponding esters during fermentation. Free amino acids can be produced by proteolysis during red wine fermentation through metabolism by endogenous enzymes or microorganisms. Microbes interact with changes in free amino acid composition. The accumulation of glutamic acid (Glu), which contributes to the freshness and umami of wine, showed a trend of significant up-regulation in the F, H and P group [47].

The corresponding discussion part has been added in Page 11 Line 49-50, as follows:

In addition, they reduce the fermentation activity of saccharomyces cerevisiae, thereby affecting the synthesis of volatile compounds of other metabolic pathways.

The corresponding discussion part has been added in Page 12 Line 5-11, as follows:

Carbohydrate metabolism and amino acid metabolism are important pathways for the production of volatile flavor compounds. The amino acids in red wine come from raw materials and microbial synthesis, and some amino acids are the precursors of higher alcohols. Therefore, the biosynthesis of amino acids and higher alcohols is related to carbohydrate metabolism and amino acid metabolism. In addition, fatty acid biosynthesis is related to carbohydrate metabolism, amino acid metabolism, and lipid metabolism [54].

The corresponding discussion part has been added in Page 12 Line 44-46, as follows:

In conclusion, the enrichment analysis of metabolic pathways in red wine samples showed that fatty acid biosynthesis, carbohydrate metabolism and amino acid metabolism play an important role in the formation of red wine flavor.

Reviewer 2 Report

This paper describes a technically sound piece of scientific research providing valuable information for understanding the potential action mechanism of pesticides on yeast growth and wine quality. Thus, it makes an important contribution to the field by assessing the effects of five pesticides (hexaconazole, difenoconazole, flutriafol, tebuconazole and propiconazole) treatment, commonly used in grape cultivation, on the growth of Saccharomyces cerevisiae strains and the produced flavor substances.

The purpose of this study is clearly presented and the literature review was critically followed. The reasons for this research are well highlighted given that only limited researches have focused on the effects of pesticide residues on microbial metabolism during the grape fermentation, although it may play an essential role in the quality and flavor of wine.

The study is well-designed and well-written, the methodology is suitable and the obtained results are convincingly presented and well aligned with the purpose of this work. They are clearly illustrated in Figures 2-8. The conclusions are presented in a concise form and clearly indicate the main findings of this study.

The cited references are relevant to the addressed research topic, but they were not properly written. Please, revise the References section according to the journal requirements (https://www.mdpi.com/journal/metabolites/instructions#preparation).

References were not cited in the text in accordance with the requirements of the journal. They should not be quoted as superscripts [1], and in the same line with the text [1]. Please, see the instructions at: https://www.mdpi.com/journal/metabolites/instructions#preparation (References) “In the text, reference numbers should be placed in square brackets [ ], and placed before the punctuation; for example [1], [1–3] or [1,3])”. 

Please, use in the text Figure 1 …., not Fig., and so on.

Through the questions addressed this study is relevant in relation to the state of art in the field and the obtained results have a high potential to move forward the knowledge in the field.

The content of this paper fits well with the journal’s scope, and therefore, it could be accepted for publication after a minor revision, to improve the quality of this work.

Author Response

Response to Reviewer 2 Comments

Point 1: he cited references are relevant to the addressed research topic, but they were not properly written. Please, revise the References section according to the journal requirements (https://www.mdpi.com/journal/metabolites/instructions#preparation).

References were not cited in the text in accordance with the requirements of the journal. They should not be quoted as superscripts [1], and in the same line with the text [1]. Please, see the instructions at: https://www.mdpi.com/journal/metabolites/instructions#preparation (References) “In the text, reference numbers should be placed in square brackets [ ], and placed before the punctuation; for example [1], [1–3] or [1,3])”.

Response 1: Thanks for your comments. Reference has been modified according to (https://www.mdpi.com/journal/metabolites/instructions#preparation).

Point 2: Please, use in the text Figure 1 …., not Fig., and so on.

Response 2: Thank you. Figure 1-8 has been changed as follows:

Fig.1 have been modified to Figure 1.

Fig.2 have been modified to Figure 2.

Fig.3 have been modified to Figure 3.

Fig.4 have been modified to Figure 4.

Fig.5 have been modified to Figure 5.

Fig.6 have been modified to Figure 6.

Fig.7 have been modified to Figure 7.

Fig.8 have been modified to Figure 8.

Reviewer 3 Report

The manuscript describes the effects of several pesticide (hexaconazole, difenoconazole, flutriafol, tebuconazole and propiconazole) on the growth of yeast. Then, the changes of flavor substance in wine were characterized by metabolomic analysis based on GC-MS.

As a consequence, the growth of yeast was significantly inhibited by these five pesticides, and the fermentation profile of yeast was significantly changed.
Several metabolites were modified in comparison of control. Most of the metabolites were natural flavor compounds, like carbohydrates, amino acids, short-chain fatty acids and alcohols.

The results of our study comprehensively analyzed the effects of pesticides on wine flavor and provided some key information for understanding the potential function mechanisms on yeast growth and wine quality.

The overall study provides a new idea for future research on the flavor quality of fermented wine.

The manuscript can be accepted, but before, it would be worthy to enlarge some of the figures. i.e. fig 5. It is difficult to understand those results if they are in the size we have now.

Congratulations

Author Response

Response to Reviewer 3 Comments

Point 1: it would be worthy to enlarge some of the figures. i.e. fig 5. It is difficult to understand those results if they are in the size we have now.

Response 1: Thanks for your comments. The picture has been adjusted clearly., and other comments have been responded to point-by-point. Please see the attachment.
